# Changes to Urinary Proteome in High-Fat-Diet *ApoE*^−/−^ Mice

**DOI:** 10.3390/biom12111569

**Published:** 2022-10-26

**Authors:** Yuanrui Hua, Wenshu Meng, Jing Wei, Yongtao Liu, Youhe Gao

**Affiliations:** Gene Engineering Drug and Biotechnology Beijing Key Laboratory, College of Life Sciences, Beijing Normal University, Beijing 100875, China

**Keywords:** ApoE^−/−^ mice, high-fat diet, urinary proteome, chemical modifications, early biomarker

## Abstract

Cardiovascular disease is currently the leading cause of death worldwide. Atherosclerosis is an important pathological basis of cardiovascular disease, and its early diagnosis is of great significance. Urine bears no need nor mechanism to be stable, so it accumulates many small changes and is therefore a good source of biomarkers in the early stages of disease. In this study, ApoE-/- mice were fed a high-fat diet for 5 months. Urine samples from the experimental group and control group (C57BL/6 mice fed a normal diet) were collected at seven time points. Proteomic analysis was used for comparison within the experimental group and for comparison between the experimental group and the control group. The results of the comparison within the experimental group showed a significant difference in the urinary proteome before and after a one-week high-fat diet, and several of the differential proteins have been reported to be associated with atherosclerosis and/or as biomarker candidates. The results of the comparison between the experimental group and the control group indicated that the biological processes enriched by the GO analysis of the differential proteins correspond to the progression of atherosclerosis. The differences in chemical modifications of urinary proteins have also been reported to be associated with the disease. This study demonstrates that urinary proteomics has the potential to sensitively monitor changes in the body and provides the possibility of identifying early biomarkers of atherosclerosis.

## 1. Introduction

Atherosclerosis (AS) is the primary pathological basis of cardiovascular disease (CVD) [1], which is the leading cause of death in the world today [2]. In 2015, more than 17 million people died of cardiovascular disease, accounting for 31% of all deaths worldwide [3]. The impact of atherosclerosis is more significant during its late stages, during which it induces a series of fatal consequences such as myocardial infarction and stroke [4]. Therefore, its early diagnosis is of vital importance.

Urine is an ideal source of early biomarkers because biomarkers are measurable changes related to biological processes regulated by homeostasis mechanisms, and urine can accumulate these early changes [5]. This conclusion has been confirmed by many related studies. For example, in a glioblastoma animal model constructed by injecting tumour cells into the brains of rats, changes in the urine proteome occurred before magnetic resonance imaging reflected the changes caused by the tumour [6]. Similarly, studies have confirmed that even if only approximately 10 cells are subcutaneously inoculated in rats, the urinary proteome can change significantly [7]. In addition, urine is more accessible and non-invasive to obtain [8].

The use of animal models avoids the influence of genetic, environmental and other factors on the urinary proteome, and it is easier to judge the early stages of atherosclerosis and identify biomarkers [9]. Apolipoprotein E (*ApoE*) plays an important role in maintaining the normal levels of cholesterol and triglycerides in serum by transporting lipids in the blood [10]. Mice lacking *ApoE* function develop hypercholesterolemia, increased very-low-density lipoprotein (VLDL) and decreased high-density lipoprotein (HDL), exhibiting spontaneous formation of plaques, and a high-fat diet can greatly accelerate the formation of plaques [11].

Although the early diagnosis of atherosclerosis is important and a large number of biomarkers have been identified, few works involve urinary biomarkers. Further, urine is an ideal source of early biomarkers and has the potential to reflect the changes in early stages of atherosclerosis. Therefore, there is an urgent need to explore the changes to the urinary proteome in high-fat-diet *ApoE^−/−^* mice.

In this experiment, *ApoE*^−/−^ mice were fed a high-fat diet for five months. At different timepoints (week 0, week 1, month 1, month 2, month 3, month 4 and month 5) of the experiment, urine samples were collected and analysed by mass spectrometry. Comparison within the experimental group was conducted, as well as comparison between the experimental group and the control group using urine samples of normal-diet C57BL/6 mice. The changes in the proteome and chemical modifications that occur during disease progression provide clues in the search for biomarkers.

## 2. Materials and Methods

### 2.1. Experimental Animals

Six 4-week-old male *ApoE*^−/−^ mice were purchased from the Laboratory Animal Science Department of Peking University Health Science Centre and fed a high-fat diet (21% fat and 0.15% cholesterol, Beijing Keao Xieli Feed Co., Ltd., Beijing, China) for 5 months. The animal licence is SCXK (Beijing) 2016-0010. A 12 h normal light–dark cycle and standard temperature (22°C ± 1°C) and humidity (65%–70%) conditions were used. All animal protocols governing the experiments in this study were approved by the Institute of Basic Medical Sciences Animal Ethics Committee, Peking Union Medical College (approved ID: ACUC-A02-2014-007). The study was carried out in compliance with the ARRIVE guidelines.

### 2.2. Histopathology

Six 6-month-old *ApoE*^−/−^ mice in the experimental group and four 6-month-old normal-diet C57BL/6 mice (purchased from Beijing Vital River Laboratory Animal Technology Co., Ltd., Beijing, China) were euthanized together. Whole arteries were dissected and stained with Oil Red O [12]. The aortas were fixed in 4% paraformaldehyde and dehydrated with isopropanol. After longitudinal incision, they were stained with Oil Red O dye solution (Biotopped, Beijing, China) for 20 min and rinsed three times with isopropanol. A digital camera (Canon, Tokyo, Japan) was then used to obtain images of the aortas, which were analysed using ImageJ software (1.52a, NIH, Bethesda, MD, USA).

### 2.3. Urine Collection and Sample Preparation

To identify the short-term effects of a high-fat diet on animals, urine samples of experimental group mice were collected during week 0 and week 1. To monitor changes in the urinary proteome during the whole process, urine samples of the experimental group at months 1, 2, 3, 4 and 5 were also collected. The urine of four C57BL/6 mice fed a normal diet (all purchased from Beijing Vital River Laboratory Animal Technology Co., Ltd., Beijing, China) corresponding to the age of *ApoE*^−/−^ mice was collected as a control group (not the same batch). All mice were placed in metabolic cages individually for 12 h to collect urine without any treatment. The collected urine samples were immediately stored at -80°C. The experimental process is shown in Figure 1.

The urine samples were centrifuged at 12,000 g for 40 min to remove the supernatant, precipitated using 3 times their volume of ethanol overnight, and then centrifuged at 12,000 g for 30 min. The protein was resuspended in lysis buffer (8 mol/L urea, 2 mol/L thiourea, 25 mmol/L dithiothreitol and 50 mmol/L Tris). The protein concentration was measured using the Bradford method. Urine proteolysis was performed using the filter-aided sample preparation (FASP) method [13]. The urine protein was loaded on the filter membrane of a 10 kDa ultrafiltration tube (PALL, Port Washington, NY, USA) and washed twice with UA (8 mol/L urea, 0.1 mol/L Tris-HCl, pH 8.5) and 25 mmol/L NH_4_HCO_3_ solution; 20 mmol/L dithiothreitol (DTT, Sigma, St. Louis, MO, USA) was added for reduction at 37°C for 1 h, and then 50 mmol/L iodoacetamide (IAA, Sigma, St. Louis, MO, USA) was used for alkylation in the dark for 30 min. After washing twice with UA and NH_4_HCO_3_ solutions, trypsin (Promega, Fitchburg, WI, USA) was added at a ratio of 1:50 for digestion at 37°C for 14 h. The peptides were passed through Oasis HLB cartridges (Waters, Milford, MA, USA) for desalting and then dried by vacuum evaporation (Thermo Fisher Scientific, Bremen, Germany).

### 2.4. Spin-Column Peptide Fractionation

The digested samples were redissolved in 0.1% formic acid and diluted to 0.5 μg/μL. Each sample was used to prepare a mixed peptide sample, and a high-pH reversed-phase fractionation spin column (Thermo Fisher Scientific, Waltham, MA, USA) was used for separation. The mixed peptide samples were added to the chromatographic column and eluted with a step gradient of 8 increasing acetonitrile concentrations (5, 7.5, 10, 12.5, 15, 17.5, 20 and 50% acetonitrile). Ten effluents were finally collected by centrifugation and were dried with vacuum evaporation and resuspended in 0.1% formic acid. In this study, iRT reagent (Biognosis, Schlieren, Switzerland) was used to calibrate the retention time of the extracted peptide peaks, which were added to ten components and each sample at a volume ratio of 10:1.

### 2.5. LC-MS/MS Analysis

An EASY-nLC 1200 chromatography system (Thermo Fisher Scientific, Waltham, MA, USA) and Orbitrap Fusion Lumos Tribrid mass spectrometer (Thermo Fisher Scientific, Waltham, MA, USA) were used for mass spectrometry acquisition and analysis. The peptide sample was loaded onto the precolumn (75 μm×2 cm, C18, 2 μm, Thermo Fisher) at a flowrate of 400 nL/min and then separated using a reversed-phase analysis column (50 μm × 15 cm, C18, 2 μm, Thermo Fisher) for 120 min. The mobile phase with a gradient of 4%–35% (80% acetonitrile + 0.1% formic acid + 20% water) was used for elution. A full MS scan was acquired within a 350–1500 m/z range with the resolution set to 120,000. The MS/MS scan was acquired in Orbitrap mode with a resolution of 30,000. The HCD collision energy was set to 30%. The mass spectrum data of 10 components separated by the reversed-phase column and all the samples obtained by enzymatic hydrolysis were collected in DDA mode.

### 2.6. Label-free DIA Quantification

The DDA collection results of the above 10 components were imported into the Proteome Discoverer software (version 2.1, Thermo Scientific, Waltham, MA, USA) search database using the following parameters: mouse database (released in 2019, containing 17,038 sequences) with the iRT peptide sequence attached, trypsin digestion, a maximum of two missing cleavage sites, parent ion mass tolerance of 10 ppm, fragment ion mass tolerance of 0.02 Da, methionine oxidation set as variable modification, cysteine carbamidomethylation set as fixed modification, and protein false discovery rate (FDR) set to 1%. The PD search result was used to establish the DIA acquisition method, and the window width and number were calculated according to the m/z distribution density.

Sixty-nine peptide samples were put into DIA mode to collect mass spectrometry data. Spectronaut™ Pulsar X (Biognosys, Biognosis, Switzerland) software was used to process and analyse mass spectrometry data [14]. Based on the DDA search result pdResult file and the 10 DDA raw files, we created a spectrum library; the raw files collected by DIA were imported for each sample to search the library. The high-confidence protein standard was a peptide q value < 0.01, and the peak area of all fragment ions of the secondary peptide was used for protein quantification.

### 2.7. Protein Chemical Modifications Search

PFind Studio software (version 3.1.6, Institute of Computing Technology, Chinese Academy of Sciences, Beijing, China) was used to perform label-free quantitative analysis of the DDA collection results of enzymatic hydrolysis samples [15]. The target search database was from the *Mus musculus* database downloaded by UniProt (updated September 2020). During the search process, the instrument type was set as HCD-FTMS, the enzyme was fully specific trypsin, and up to 2 missed cleaved sites were allowed. The “open-search” mode was selected, and the screening condition was that the FDR at the peptide level was less than 1%. The data were analysed using both forward and reverse database search strategies. After the initial screening, a restricted search method was used for verification.

### 2.8. Statistical Analysis

The missing abundance values were determined (KNN method) [16], and CV value screening (CV < 0.3) [17] was performed on the mass spectrometry results. The two-sided unpaired t-test was used for the comparison between each set of data. Comparison within the experimental group and comparison between the experimental group and the control group at the same time points were screened for differential proteins. The screening criteria were as follows: fold change (FC) between the two groups ≥1.5 or ≤0.67 and *p <* 0.05. At the same time, the samples in each two groups were randomly combined, and the average number of differential proteins in all permutations and combinations was calculated according to the same criteria as normal screening (Appendix A), ensuring that differential proteins were generated by differences between groups rather than random production.

The proportions of different types of chemical modification sites out of the total number of modification sites were calculated, and the data between each two groups were compared by two-sided unpaired t-tests. The screening criteria were FC between the two groups ≥1.5 or ≤0.67 and *p <* 0.05.

The DAVID database (https://url.cy/0E13rJ) [18] was used to perform functional enrichment analysis on the differential proteins that were screened. The significance threshold of *p <* 0.05 was adopted. All methods were performed in accordance with the relevant guidelines and regulations.

## 3. Results

### 3.1. Histopathology

The Oil Red O staining results of the whole aortas of 6-month-old *ApoE*^−/−^ mice fed a high-fat diet for 5 months were compared to those of 6-month-old mice fed a normal diet. The average percentage of stained areas in the experimental group was 17.78 ± 2.14% (*n* = 6), and the average percentage in the control group was 0.88 ± 0.34% (*n* = 4), *p* = 0.0004 (Figure 2).

### 3.2. Differential Protein Screening and Functional Annotation

The experimental group and the control group had 69 samples from seven time points (W0/W1/M1/M2/M3/M4/M5) for non-labelled LC-MS/MS quantification (one sample in the experimental group for W0 was insufficient). A total of 592 proteins identified with at least 2 unique peptides with FDR < 1% were identified, and an average of 360 urine proteins were identified for each sample. The heatmap (Appendix A) of all the samples shows that it is hard to discriminate samples of different time points or groups as a whole, which indicates that there are great differences among individuals. The mass spectrometry proteomics data have been deposited to the ProteomeXchange Consortium (https://url.cy/qevTk1 (accessed on 10 August 2022)) via the iProX partner repository [19] with the dataset identifier PXD027610.

#### 3.2.1. Comparison within the Experimental Group

##### Short-Term Effects of a High-Fat Diet

To identify the effects of a high-fat diet, after a week of a high-fat diet in *ApoE*^−/−^ mice, urine samples collected from W0 and W1 were compared and analysed. The volcano plot of proteins is shown in Appendix A. A total of 12 proteins were significantly upregulated and 15 proteins were significantly downregulated at W1 (Table 1). Among them, 21 proteins or their family members have been reported to be associated with lipids.

GO analysis of these 27 proteins by DAVID showed that most of the annotated biological processes were related to lipid metabolism and glucose metabolism (Figure 3). At the same time, the differential proteins between W1 and W0 in the control group (Appendix A) did not enrich for any significant changes in biological processes, indicating that the physiological state of mice did not change significantly at W1, while only a week of a high-fat diet induced huge changes in the animal urinary proteome, further demonstrating that the urinary proteome sensitively reflects changes in the body.

##### Urinary Proteome Changes in the Whole Process

Compared to W0, 51/69/86/65/88 proteins changed significantly at M1/M2/M3/M4/M5 in the experimental group, respectively. The volcano plots of proteins are shown in Appendix A. The Venn diagram (Figure 4) shows that a total of 17 proteins changed significantly at all five time points, and the DIA quantitative results show that these 17 proteins exhibited the same change trend at these time points. Another 18 proteins changed significantly at the last four time points, and the trend of change was the same at each time point (Appendix A). Among them, 26 proteins or their family members have been previously reported to be related to lipid metabolism or cardiovascular diseases.

Major urinary proteins (MUPs) are members of the lipocalcin family, which can be isolated and transport various lipophilic molecules in the blood and other body fluids [20]. Knockout of mouse trefoil factor 2 protects against obesity in response to a high-fat diet [21]. Angiotensinogen plays a key role in fat cell metabolism and inflammation development [22]. Alpha1-antitrypsin has been reported as a biomarker of atherosclerosis [23]. It has been reported that CCN4 (cellular communication network factor 4) promotes the migration and proliferation of vascular smooth muscle cells by interacting with α5β1 integrin [24], which plays a vital role in the occurrence and development of atherosclerosis. Regular monitoring of vitamin B12 status may help prevent atherosclerosis-related diseases, and anticobalamin 2 can carry vitamin B12 [25]. Regenerated islet-derived protein 3β, an inflammatory marker, is of great significance for the recruitment of macrophages and for tissue repair [26]. The level of α-2-HS-glycoprotein is positively correlated with atherosclerotic substitution parameters, such as intima–media thickness (IMT) and arteriosclerosis [27]. The literature shows that gelsolin stabilizes actin filaments by binding to the ends of filaments, preventing monomer exchange. Its downregulation indicates that the cytoskeleton of vascular smooth muscle cells in the human coronary atherosclerotic medium is dysregulated [28]. It has been reported that *SCUBE2* may play an important role in the progression of atherosclerotic plaques through Hh signal transduction [29]. Type I collagen is an early biomarker of atherosclerosis [23].

Igκ chain V-III region PC 7043, Igκ chain V-II region 26–10 and immunoglobulin κ constant are all involved in the adaptive immune response. The haptoglobin polymorphism is related to the prevalence and clinical evolution of many inflammatory diseases, including atherosclerosis [30]. MHCII antigen presentation has an important protective function in atherosclerosis [31]. Interleukin-18 plays a key role in atherosclerosis and plays a role in appetite control and the development of obesity [32]. According to the literature, compared to healthy controls, *LAMP-2* gene expression and protein levels in peripheral blood leukocytes of patients with coronary heart disease are significantly increased [33]. T-cadherin is essential for the accumulation of adiponectin in neointima and atherosclerotic plaque lesions [34]. Kidney androgen-regulated protein has also been reported in the urine of *ApoE*^−/−^ mice fed a high-fat diet [23]. Fibronectin is an indicator of connective tissue formation during atherosclerosis [35]. Peripheral arterial occlusive disease (PAOD) is one of the primary manifestations of systemic atherosclerosis, and transthyretin and complement factor B are potential markers for monitoring plasma PAOD disease [36]. Serotransferrin plays an important role in atherosclerosis [37]. The differential expression of serine protease inhibitor A3 in blood vessels is significantly related to human atherosclerosis [38]. Prolactin plays a role in the proliferation of vascular smooth muscle cells, and the proliferation of vascular smooth muscle cells is a characteristic of cardiovascular diseases such as hypertension and atherosclerosis [39].

The abovementioned differential proteins that continually changed during the whole process were analysed using DAVID for GO analysis (Figure 4), and the enriched biological processes are also shown in the figure.

The major urinary protein-induced lipid metabolism- and glucose metabolism-related biological processes changed significantly; the acute phase reaction has been reported in the literature to be related to atherosclerosis [40]. The positive regulation of fibroblast proliferation also changed significantly, and vascular damage and dysfunction of adipose tissue around blood vessels promotes vasodilation, fibroblast activation and myofibroblast differentiation [41]. Wound healing is also related to atherosclerosis [42]. The extracellular matrix gives atherosclerotic lesion areas tensile strength, viscoelasticity, and compressibility [43]. There are also reports showing correlation between osteoporosis and atherosclerosis [44]. The ERK1/ERK2 pathway is involved in insulin (INS) and thrombin-induced vascular smooth muscle cells, which play important roles in proliferation [45]. Cell adhesion also plays an important role in atherosclerosis [46].

The comparison within the experimental group avoids the influence of genetic and dietary differences on the experimental results to the greatest extent, but the influence of biological growth and development is difficult to avoid. The results show that there are a variety of proteins that change continually throughout the progression of the disease and that are closely related to the disease. It is worth noting that the differential proteins obtained using this comparison method and the biological processes and pathways enriched by them exhibit a high degree of overlap at different time points, which may make it difficult to enhance early diagnosis of the disease, so follow-up comparison between the experimental group and control group was performed.

#### 3.2.2. Comparison between the Experimental Group and the Control Group

Comparison of the results between the experimental group and the control group at the same time points showed that 44/16/54/23/48/57/46 differential proteins were obtained at W0/W1/M1/M2/M3/M4/M5, respectively. The details of the proteins are shown in Table 2, the volcano plots of proteins are shown in Appendix A, and the overlap of differential proteins at different time points is shown in Appendix A. Comparing between the experimental group and control group, there were significant differences in the differential proteins at each time point, but they were all closely related to lipids and cardiovascular diseases.

The differential proteins were analysed by DAVID for GO analysis, and the biological processes that changed significantly at different time points are shown in Figure 5. The biological processes related to lipid metabolism and glucose metabolism in the experimental group were significantly different from those in the control group at W0. At W0 and M4, the immune-related processes were significantly different. Differential proteins at M1 were primarily enriched in cell adhesion-related processes, while at M2, they were primarily enriched in redox reaction-related processes. At M3, wound healing began to appear, and there were many adhesion-related processes. In addition to a large number of immune-related processes, the positive regulation of fibroblast proliferation and the negative regulation of angiogenesis also appeared at M4. The processes related to phagocytosis and proteolysis began to appear at M5.

##### Effects of Genetic Factors

At W0, before a high-fat diet was administered to the experimental group, the only difference between the two groups was genetic factors. There were already significant differences in the biological processes related to lipid and glycometabolism, indicating that *ApoE* gene knockout greatly affects the lipid transport in mice in the experimental group, which is reflected by the urinary proteome very early. Acute phase reactions, immune responses, cytokines and proteolysis are also closely related to atherosclerosis [47,48,49,50].

##### Urinary Proteome Changes during Whole Process

The literature shows that during the early stages of atherosclerosis, low-density lipoprotein (LDL) particles accumulate in the arterial intima, and are thereby protected from plasma antioxidants and undergo oxidation and other modifications and have proinflammatory and immunogenic properties. Classic monocytes circulating in the blood can exhibit anti-inflammatory functions and bind to the adhesion molecules expressed by activated endothelial cells to enter the inner membrane. Once in the inner membrane, monocytes can mature into macrophages, which express scavenger receptors that bind to lipoprotein particles and then become foam cells, finally forming the core of atherosclerotic plaques. T lymphocytes can also enter the inner membrane to regulate the functions of natural immune cells, endothelial cells and smooth muscle cells. The smooth muscle cells in the media can migrate to the inner membrane under the action of leukocytes to secrete extracellular matrix and form a fibrous cap [51]. During the exploration of this experiment, at week 1, the differentially expressed proteins between the experimental and control groups were related to the differentiation of epithelial cells, and cell adhesion was enriched in M1 macrophages, which may be related to the adhesion of monocytes. Differential proteins between the experimental and control groups at M2 were related to biological processes associated with redox, which may be related to the redox of LDL particles. Cell adhesion also changes at M3, which may involve the recruitment of phagocytes. Numerous immune-related biological processes changed in M4, indicating the participation of immune cells such as T cells. The regulation of fibroblast proliferation may be related to the formation of fibrous caps. Enriched results revealed that proteolysis changed significantly at M5. It has been reported that activated macrophages can secrete proteolytic enzymes and degrade matrix components. The loss of matrix components may subsequently lead to plaque instability and increase the risk of plaque rupture and thrombosis [52]. Fibrin dissolution also plays an important role in the development of atherosclerosis [53].

The biological processes of the enrichment of differential proteins at different time points can correspond to the occurrence and development of atherosclerosis, indicating that the urinary proteome has the potential to be used to monitor the disease process.

After a week of a high-fat diet in the experimental group, the protein kinase B signalling pathway changed. It has been reported to play an important role in the survival, proliferation and migration of macrophages and may affect the development of atherosclerosis [54]. After a month of a high-fat diet, many biological processes underwent significant changes. Studies have shown that urinary sodium excretion is the decisive factor in carotid intima–media thickness, which is an indicator of atherosclerosis [55]. The classical pathway of complement activation is also related to atherosclerosis [56]. Copper and isotypic cysteine can interact to generate free radicals, thereby oxidizing LDL, which has been found in atherosclerotic plaques [57]. At M2, oestrogen is also reported to have a variety of anti-atherosclerotic properties, including affecting plasma lipoprotein levels and stimulating the production of prostacyclin and nitric oxide [58]. At M3, wound healing is also associated with atherosclerosis [42]. For the biological processes that changed at M4, the ERK1/ERK2 pathway plays an important role in the proliferation of vascular smooth muscle cells induced by insulin (INS) and thrombin [45]. Alternative pathways of complement activation and major histocompatibility complex family II have been reported to be associated with atherosclerosis [59,60]. In the enrichment of differential proteins at M5, chaperone-mediated autophagy (CMA) plays an important upstream regulatory role in lipid metabolism [61].

To further explore the effect of high-fat diet on chemical modifications of urine proteins, a total of 15 samples were selected at three time points (EW0/EM5/CM5). After data retrieval (.raw) based on open-pFind software, the analysis results were exported in pBuild.

A total of 923 different chemical modification types were identified in 15 samples, of which 468 chemical modification types were identified in the EW0 group, 748 chemical modification types were identified in the EM5 group, and 611 chemical modification types were identified in the CM5 group.

An unsupervised cluster analysis of all modifications found that the CM5 group was well distinguished from the other two groups (Figure 6). The percentages of different modification types in the EW0 group and the EM5 group were quantified to identify the modification changes that occurred in the comparison within the experimental group. Among them, one modification type was unique to the EW0 group and existed in more than four samples (the total number of samples was five), 23 modification types were unique to the EM5 group and existed in more than five samples (the total number of samples was six); there are 68 types of modifications shared by the two groups, and there had significant differences (FC ≥ 1.5 or ≤0.67, *p <* 0.05). At the same time, the proportions of different types of modified sites in the CM5 group and the EM5 group were quantified, and the difference between the experimental group and the control group was analysed. Among them, eight modification types were unique to the CM5 group and existed in more than three samples (the total number of samples was four), and 19 modification types were unique to the EM5 group and existed in more than five samples (the total number of samples was six). There were 72 types of modifications that were shared by the two groups that had significant differences (FC ≥ 1.5 or ≤0.67, *p <* 0.05) (see Appendix A for details).

To reduce the false negative influence caused by the open search mode, a restricted search method was used for verification. Modification types that accounted for the top five modification sites in the open search were fixed; modification types that were unique in a group and existed in each sample and modification types that had been reported related to lipids in the literature were selected. Twenty modifications in the EM5-EW0 group and 25 modifications in the EW5-CM5 group were selected, and the proportion of modified sites in the total number of sites was calculated (Appendix A). The screening criteria were FC ≥ 1.5 or ≤ 0.67 and *p <* 0.05. Finally, in the comparison within the experimental group (EM5-EW0), N-terminal carbamylation (Carbamyl[AnyN-term]), CHDH modification of aspartic acid (CHDH[D]), tryptophan to kynurenine acid substitution (Trp- > Kynurenin[W]), oxidation modification of proline (Oxidation[P]), cysteine modification of cysteine (Cysteinyl[C]), sulphur dioxide modification of cysteine (SulfurDioxide[C]), NO_SMX_SIMD modification of cysteine (NO_SMX_SIMD[C]) and Delta_H(2)C(3) modification of lysine (Delta_H(2)C(3)[K]) significantly changed. In the comparison between the experimental and control groups (EM5–CM5), guanidine modification of lysine (Guanidinyl[K]), phosphouridine modification of tyrosine (PhosphoUridine[Y]), N-terminal carbamoyl modification (Carbamyl[AnyN-term]), Delta_H(2)C(2) modification (Delta_H(2)C(2)[AnyN-term]) at the N-terminus and Dihydroxyimidazolidine modification of arginine (Dihydroxyimidazolidine) [R]) showed significant changes.

**Table 1 biomolecules-12-01569-t001:** Details of differential proteins between week 1 and week 0 samples in the experimental group.

UniProt	Human UniProt	Protein Name	*p*-Value	Fold Change	References
B5X0G2	No	Major urinary protein 17	0.0008	7.92	[20]
P11588	No	Major urinary protein 1	0.0007	7.84	[20]
A2BIM8	No	Major urinary protein 18	0.0014	3.19	[20]
Q9JI02	No	Secretoglobin family 2B member 20	0.0488	2.75	—
Q5FW60	No	Major urinary protein 20	0.0121	2.65	[20]
Q07797	Q08380	Galectin-3-binding protein	0.0077	2.59	[62,63]
Q61838	No	Pregnancy zone protein	0.0411	2.46	[64]
P11591	No	Major urinary protein 5	0.0136	2.40	[20]
Q64695	Q9UNN8	Endothelial protein C receptor	0.0150	2.23	[65]
Q91WR8	P59796	Glutathione peroxidase 6	0.0469	1.99	[66]
P06797	P07711	Cathepsin L1	0.0146	1.84	[67]
P13597	P05362	Intercellular adhesion molecule 1	0.0432	1.66	[68,69]
Q9JK39	A8MVZ5	Butyrophilin-like protein 10	0.0446	0.60	—
P01898	P01891	H-2 class I histocompatibility antigen, Q10 alpha chain	0.0429	0.59	[70]
P55292	Q02487	Desmocollin-2	0.0160	0.57	[71,72]
P23780	P16278	Beta-galactosidase	0.0384	0.56	—
Q60648	P17900	Ganglioside GM2 activator	0.0382	0.56	[73]
P00688	P04746	Pancreatic alpha-amylase	0.0311	0.55	[74]
P70269	P14091	Cathepsin E	0.0299	0.54	[75]
P11859	P01019	Angiotensinogen	0.0338	0.51	[22]
Q6UGQ3	No	Secretoglobin family 2B member 2	0.0270	0.49	—
O88322	Q14112	Nidogen-2	0.0004	0.43	—
O88968	P20062	Transcobalamin-2	0.0155	0.39	[25]
P11087	P02452	Collagen alpha-1(I) chain	0.0080	0.34	[23]
Q4KML4	Q9P1F3	Costars family protein ABRACL	0.0181	0.28	—
P35230	Q06141	Regenerating islet-derived protein 3-beta	0.0012	0.24	[26]
A2AEP0	No	Odorant-binding protein 1b	0.0213	0.20	[76]

**Table 2 biomolecules-12-01569-t002:** Details of differential proteins between the experimental group and the control group at different time points.

UniProt	Human UniProt	Protein Name	Fold Change							References
			EW0-CW0	EW1-CW1	EM1-CM1	EM2-CM2	EM3-CM3	EM4-CM4	EM5-CM5	
P35230	Q06141	Regenerating islet-derived protein 3-beta	5.22	—	—	—	—	—	—	[26]
P13020	P06396	Gelsolin	2.83	—	—	—	—	—	—	[28]
P97426	P12724	Eosinophil cationic protein 1	2.57	—	—	—	1.77	7.57	—	[77]
O88322	Q14112	Nidogen-2	2.32	1.96	—	—	—	—	0.39	—
P29699	P02765	Alpha-2-HS-glycoprotein	2.27	—	—	—	—	—	—	[27]
P07758	P01009	Alpha-1-antitrypsin 1-1	2.11	—	—	—	0.58	0.25	—	[23]
P07309	P02766	Transthyretin	2.06	—	—	—	—	—	—	[78]
P49183	P24855	Deoxyribonuclease-1	1.79	—	0.47	—	—	—	—	[79]
P01864	No	Ig gamma-2A chain C region secreted form	1.55	—	0.30	—	—	—	—	—
P19221	P00734	Prothrombin	0.63	—	—	—	—	—	0.54	[80]
P61110	P61109	Kidney androgen-regulated protein	0.62	—	—	—	—	6.55	—	[23]
Q9Z0M9	O95998	Interleukin-18-binding protein	0.60	—	—	—	2.11	—	—	[32]
P05533	No	Lymphocyte antigen 6A-2/6E-1	0.60	—	—	—	—	—	—	—
O09043	O96009	Napsin-A	0.59	—	—	—	0.42	—	—	—
P03953	P00746	Complement factor D	0.59	—	—	—	—	2.60	—	[81]
Q91VW3	Q9H299	SH3 domain-binding glutamic acid-rich-like protein 3	0.58	—	—	2.22	—	—	—	—

P15379	P16070	CD44 antigen	0.58	—	0.25	—	—	3.43	—	[82]
P04441	P04233	H-2 class II histocompatibility antigen gamma chain	0.54	—	—	—	4.59	13.02	—	[31]

P25119	P20333	Tumour necrosis factor receptor superfamily member 1B	0.54	—	—	—	1.74	—	—	[83]

Q00993	P30530	Tyrosine-protein kinase receptor UFO	0.53	—	—	0.51	2.16	3.62	—	[84]
P07361	P02763	Alpha-1-acid glycoprotein 2	0.52	—	—	—	—	0.13	—	[85]
P09470	P12821	Angiotensin-converting enzyme	0.51	—	—	—	—	—	—	[86]
Q91WR8	P59796	Glutathione peroxidase 6	0.47	—	—	—	—	3.34	—	[66]
Q62395	Q07654	Trefoil factor 3	0.46	—	—	—	—	—	—	[21]
Q9DAK9	Q9NRX4	14 kDa phosphohistidine phosphatase	0.46	—	—	—	—	—	2.02	[87]
O88188	O95711	Lymphocyte antigen 86	0.45	—	—	—	—	—	—	—
Q60932	P21796	Voltage-dependent anion-selective channel protein 1	0.44	0.45	—	3.68	—	—	—	[88]

P11589	No	Major urinary protein 2	0.44	—	3.82	—	1.98	6.19	3.16	[20]
Q62266	No	Cornifin-A	0.43	0.49	—	—	—	—	—	—
P17047	P13473	Lysosome-associated membrane glycoprotein 2	0.42	—	—	—	2.19	3.45	2.44	[33]
P0CW03	No	Lymphocyte antigen 6C2	0.41	—	—	—	—	—	—	—
Q60590	P02763	Alpha-1-acid glycoprotein 1	0.41	—	—	—	—	0.08	—	[85]
P01665	No	Ig kappa chain V-III region PC 7043	0.41	—	—	—	—	7.91	2.38	—
P04939	No	Major urinary protein 3	0.39	—	—	—	—	—	—	[20]
Q6SJQ5	Q6UXZ3	CMRF35-like molecule 3	0.35	—	—	—	—	—	0.56	—
Q64695	Q9UNN8	Endothelial protein C receptor	0.31	—	—	—	—	—	—	[65]
E9Q557	P15924	Desmoplakin	0.29	—	—	—	3.82	—	—	—
P11591	No	Major urinary protein 5	0.27	—	—	—	—	4.49	2.30	[20]
P51437	P49913	Cathelicidin antimicrobial peptide	0.24	—	—	—	—	—	—	[89]
Q5FW60	No	Major urinary protein 20	0.23	—	—	—	—	—	2.14	[20]
A2BIM8	No	Major urinary protein 18	0.20	0.50	6.10	—	2.10	—	—	[20]
Q61646	P00738	Haptoglobin	0.14	—	—	—	—	0.09	—	[30]
P11588	No	Major urinary protein 1	0.07	0.45	—	—	2.17	—	—	[20]
B5X0G2	No	Major urinary protein 17	0.06	0.34	28.79	—	—	—	4.06	[20]
Q9JI02	No	Secretoglobin family 2B member 20	—	2.83	—	—	0.23	—	—	—
Q01279	P00533	Epidermal growth factor receptor	—	1.52	—	—	—	—	—	[90]
P10605	P07858	Cathepsin B	—	0.63	0.44	—	—	—	—	[91]
P50429	P15848	Arylsulfatase B	—	0.56	—	—	—	—	—	[92]
Q571E4	P34059	N-acetylgalactosamine-6-sulfatase	—	0.52	—	—	—	3.57	—	—
Q9JK39	A8MVZ5	Butyrophilin-like protein 10	—	0.50	—	—	—	—	—	—
P23780	P16278	Beta-galactosidase	—	0.48	0.15	—	0.28	—	—	—
P70269	P14091	Cathepsin E	—	0.42	—	—	—	2.14	—	[75]
O35887	O43852	Calumenin	—	0.40	—	—	2.88	4.38	1.88	[93]
Q9EP95	Q9BQ08	Resistin-like alpha	—	0.27	—	—	—	—	0.20	[94]
P20152	P08670	Vimentin	—	—	16.24	—	—	—	—	[95]
Q8K0E8	P02675	Fibrinogen beta chain	—	—	8.28	—	—	—	—	[96]
P16858	P04406	Glyceraldehyde-3-phosphate dehydrogenase	—	—	4.72	—	—	—	—	[97]
Q9WTR5	P55290	Cadherin-13	—	—	2.78	—	—	—	—	[34]
Q00897	P01009	Alpha-1-antitrypsin 1-4	—	—	2.57	—	—	—	3.27	[23]
O09164	P08294	Extracellular superoxide dismutase [Cu–Zn]	—	—	2.11	1.56	—	3.37	—	[98]
P11276	P02751	Fibronectin	—	—	0.58	—	—	2.15	—	[35]
Q9Z0J0	P61916	NPC intracellular cholesterol transporter 2	—	—	0.57	—	—	—	—	[99]
Q8BPB5	Q12805	EGF-containing fibulin-like extracellular matrix protein 1	—	—	0.53	0.46	—	—	—	—

Q8BZT5	Q9H756	Leucine-rich repeat-containing protein 19	—	—	0.52	0.50	0.55	—	—	[100]
P21614	P02774	Vitamin D-binding protein	—	—	0.50	—	—	—	0.49	[101]
P16675	P10619	Lysosomal protective protein	—	—	0.49	—	—	—	2.51	—
Q61147	P00450	Ceruloplasmin	—	—	0.49	—	—	0.35	—	[102]
P23953	No	Carboxylesterase 1C	—	—	0.48	—	—	—	—	—
Q61398	Q15113	Procollagen C-endopeptidase enhancer 1	—	—	0.48	—	—	—	—	[103]
O35664	P48551	Interferon alpha/beta receptor 2	—	—	0.47	—	—	—	—	[104]
P11859	P01019	Angiotensinogen	—	—	0.45	—	—	0.15	—	[22]
P01898	P01891	H-2 class I histocompatibility antigen, Q10 alpha chain	—	—	0.45	—	—	—	2.03	[70]

C0HKG5	No	Ribonuclease T2-A	—	—	0.42	0.60	—	—	—	—
Q61271	P36896	Activin receptor type-1B	—	—	0.42	—	—	3.64	—	—
O88968	P20062	Transcobalamin-2	—	—	0.40	—	—	—	—	[25]
Q9Z0L8	Q92820	Gamma-glutamyl hydrolase	—	—	0.39	—	—	—	—	[105,106]
P35459	Q14210	Lymphocyte antigen 6D	—	—	0.39	0.57	—	—	—	—
Q61129	P05156	Complement factor I	—	—	0.38	—	—	—	—	—
P01878	No	Ig alpha chain C region	—	—	0.38	—	5.03	—	—	—
P55292	Q02487	Desmocollin-2	—	—	0.38	—	3.72	—	—	[71,72]
Q9JJS0	Q9NQ36	Signal peptide, CUB and EGF-like domain-containing protein 2	—	—	0.35	—	—	—	—	[29]

Q9Z319	Q9Y5Q5	Atrial natriuretic peptide-converting enzyme	—	—	0.35	—	—	—	—	[107]
P09036	P00995	Serine protease inhibitor Kazal-type 1	—	—	0.34	—	—	—	—	—
Q4KML4	Q9P1F3	Costars family protein ABRACL	—	—	0.33	—	2.18	—	—	—
Q925F2	Q96AP7	Endothelial cell-selective adhesion molecule	—	—	0.32	—	—	—	0.48	[108]
O89020	P43652	Afamin	—	—	0.31	0.50	—	—	—	[109]
Q9DAU7	Q14508	WAP four-disulfide core domain protein 2	—	—	0.31	—	—	—	—	—
Q8BND5	O00391	Sulfhydryl oxidase 1	—	—	0.30	0.40	—	—	0.61	—
P09803	P12830	Cadherin-1	—	—	0.29	—	2.78	2.34	—	[30]
P02816	P12273	Prolactin-inducible protein homolog	—	—	0.27	—	0.45	—	—	[39]
Q91WR6	Q9NU53	Glycoprotein integral membrane protein 1	—	—	0.26	0.59	—	—	0.45	—
Q3UDR8	Q9GZM5	Protein YIPF3	—	—	0.26	—	—	—	—	—
Q6UGQ3	No	Secretoglobin family 2B member 2	—	—	0.25	—	—	—	2.03	—
Q9D3H2	No	Odorant-binding protein 1a	—	—	0.25	—	1.75	—	—	[76]
P20060	P07686	Beta-hexosaminidase subunit beta	—	—	0.23	—	0.35	—	—	—
Q8K1H9	Q9NY56	Odorant-binding protein 2a	—	—	0.21	—	0.31	0.21	—	[76]
A2AEP0	No	Odorant-binding protein 1b	—	—	0.20	—	—	—	—	[76]
Q8C6C9	Q6P5S2	Protein LEG1 homolog	—	—	0.15	—	0.44	—	—	—
P00688	P04746	Pancreatic alpha-amylase	—	—	0.14	—	—	0.32	—	[74]
P10287	P22223	Cadherin-3	—	—	0.13	—	3.19	5.62	—	[30]
P56386	P60022	Beta-defensin 1	—	—	—	3.94	—	5.25	1.90	[110]
P10639	P10599	Thioredoxin	—	—	—	3.63	—	—	—	[111]
O88844	O75874	Isocitrate dehydrogenase [NADP] cytoplasmic	—	—	—	2.71	—	—	—	[112]
Q00623	P02647	Apolipoprotein A-I	—	—	—	2.07	—	—	0.36	[113]
P0CG49	P0CG47	Polyubiquitin-B	—	—	—	1.98	—	—	—	[114]
Q03404	Q03403	Trefoil factor 2	—	—	—	1.69	2.11	8.94	—	[21]
P15947	P06870	Kallikrein-1	—	—	—	0.66	—	1.74	—	[115]
O55186	P13987	CD59A glycoprotein	—	—	—	0.61	—	2.53	—	[116]
Q921I1	P02787	Serotransferrin	—	—	—	0.57	0.43	0.09	0.21	[37]
Q60648	P17900	Ganglioside GM2 activator	—	—	—	0.51	—	—	2.27	[74]
O88792	Q9Y624	Junctional adhesion molecule A	—	—	—	0.49	—	—	—	[117]
P07724	P02768	Albumin	—	—	—	0.38	—	0.38	0.43	[118]
O70554	No	Small proline-rich protein 2B	—	—	—	—	7.98	—	3.26	[119,120]
Q62267	No	Cornifin-B	—	—	—	—	5.69	—	—	—
P35700	Q06830	Peroxiredoxin-1	—	—	—	—	3.75	—	—	[121]
P18761	P23280	Carbonic anhydrase 6	—	—	—	—	3.37	—	—	[122]
P01631	No	Ig kappa chain V-II region 26-10	—	—	—	—	3.29	3.00	—	—
P11087	P02452	Collagen alpha-1(I) chain	—	—	—	—	3.21	—	—	[23]
P01837	P01834	Immunoglobulin kappa constant	—	—	—	—	3.19	—	—	—
Q99N23	No	Carbonic anhydrase 15	—	—	—	—	2.91	—	—	[122]
P10126	P68104	Elongation factor 1-alpha 1	—	—	—	—	2.90	—	—	[123]
P70663	Q14515	SPARC-like protein 1	—	—	—	—	2.74	—	—	—
Q60847	Q99715	Collagen alpha-1(XII) chain	—	—	—	—	2.68	—	—	[23]
P29533	P19320	Vascular cell adhesion protein 1	—	—	—	—	2.62	2.12	—	[124]
O55135	P56537	Eukaryotic translation initiation factor 6	—	—	—	—	2.55	—	—	—
O54775	O95388	CCN family member 4	—	—	—	—	2.47	—	0.47	[24]
P01843	No	Ig lambda-1 chain C region	—	—	—	—	2.45	—	—	—
Q9DBV4	Q9BRK3	Matrix remodelling-associated protein 8	—	—	—	—	1.70	—	1.80	—
O35608	O15123	Angiopoietin-2	—	—	—	—	1.61	—	—	[125]
Q07797	Q08380	Galectin-3-binding protein	—	—	—	—	1.59	—	—	[62]
Q04519	P17405	Sphingomyelin phosphodiesterase	—	—	—	—	0.57	—	1.89	[126]
Q61838	No	Pregnancy zone protein	—	—	—	—	0.52	—	—	[64]
Q08423	P04155	Trefoil factor 1	—	—	—	—	—	12.79	—	[21]
O08997	O00244	Copper transport protein ATOX1	—	—	—	—	—	8.07	—	[127]
P03977	No	Ig kappa chain V-III region 50S10.1	—	—	—	—	—	4.79	—	—
P42567	P42566	Epidermal growth factor receptor substrate 15	—	—	—	—	—	4.53	—	[90]
Q91X17	P07911	Uromodulin	—	—	—	—	—	4.22	—	[128]
P57096	O43653	Prostate stem cell antigen	—	—	—	—	—	3.53	—	—
P70699	P10253	Lysosomal alpha-glucosidase	—	—	—	—	—	3.37	—	[129]
P01132	P01133	Pro-epidermal growth factor	—	—	—	—	—	3.13	—	—
P32507	Q92692	Nectin-2	—	—	—	—	—	3.03	—	[130]
Q5SSE9	Q86UQ4	ATP-binding cassette sub-family A member 13	—	—	—	—	—	2.92	—	—
P06797	O60911	Cathepsin L1	—	—	—	—	—	2.63	—	[67,75]
Q8R242	Q01459	Di-N-acetylchitobiase	—	—	—	—	—	2.47	—	—
Q9Z0K8	O95497	Pantetheinase	—	—	—	—	—	2.32	3.65	—
O54782	Q9Y2E5	Epididymis-specific alpha-mannosidase	—	—	—	—	—	1.99	—	—
P22599	P01009	Alpha-1-antitrypsin 1-2	—	—	—	—	—	0.50	—	[23]
P13634	P00915	Carbonic anhydrase 1	—	—	—	—	—	0.39	3.37	[122]
P20918	P00747	Plasminogen	—	—	—	—	—	0.32	0.63	[131]
P07759	P01011	Serine protease inhibitor A3K	—	—	—	—	—	0.26	—	[38]
P02088	P68871	Hemoglobin subunit beta-1	—	—	—	—	—	0.24	—	—
P01027	P01024	Complement C3	—	—	—	—	—	0.20	—	[56]
Q00898	P01009	Alpha-1-antitrypsin 1-5	—	—	—	—	—	0.18	—	[23]
P11672	P80188	Neutrophil gelatinase-associated lipocalin	—	—	—	—	—	0.04	—	[132]
P01887	P61769	Beta-2-microglobulin	—	—	—	—	—	—	3.07	[133]
O09114	P41222	Prostaglandin-H2 D-isomerase	—	—	—	—	—	—	2.41	[134]
Q9WUU7	Q9UBR2	Cathepsin Z	—	—	—	—	—	—	2.32	[135]
Q8BHC0	Q9Y5Y7	Lymphatic vessel endothelial hyaluronic acid receptor 1	—	—	—	—	—	—	0.61	—

O70570	P01833	Polymeric immunoglobulin receptor	—	—	—	—	—	—	0.55	—
P26041	P26038	Moesin	—	—	—	—	—	—	0.53	[136]
Q7TMJ8	Q96FE7	Phosphoinositide-3-kinase-interacting protein 1	—	—	—	—	—	—	0.52	[137]
P01660	No	Ig kappa chain V-III region PC 3741/TEPC 111	—	—	—	—	—	—	0.43	—
Q60928	P19440	Glutathione hydrolase 1 proenzyme	—	—	—	—	—	—	0.39	[106]
P61971	P61970	Nuclear transport factor 2	—	—	—	—	—	—	0.36	—
P06330	No	Ig heavy chain V region AC38 205.12	—	—	—	—	—	—	0.36	—
Q921W8	Q8WVN6	Secreted and transmembrane protein 1A	—	—	—	—	—	—	0.35	—
Q8BX43	Q969Z4	Tumour necrosis factor receptor superfamily member 19L	—	—	—	—	—	—	0.33	[83]
		Proteoglycan 4								
Q9JM99	Q92954		—	—	—	—	—	—	0.26	[137]

Among the changes observed in the comparison within the experimental group, many studies have shown that carbamylated proteins are involved in the occurrence of diseases, especially atherosclerosis and chronic renal failure [138]. The kynurenine pathway is the primary pathway of tryptophan metabolism and plays an important role in early atherosclerosis [139]. The oxidation of proline can form glutamate semialdehyde, and glutamate semialdehyde is closely related to lipid peroxidation [140]. Elevated plasma homocysteine has also been widely studied as an independent risk factor for atherosclerosis [141]. Obstruction of the sulphur dioxide/aspartate aminotransferase pathway is also known to be involved in the pathogenesis of many cardiovascular diseases [142]. The Delta_H(2)C(3) modification of lysine also refers to acrolein addition +38, and acrolein and other α- and β-unsaturated aldehydes are considered to be mediators of inflammation and vascular dysfunction [143]. CHDH modification of aspartic acid and NO_SMX_SIMD modification of cysteine have not been reported to be related to atherosclerosis and may act as potential modification sites.

Although it was not verified in a restricted search, there are also studies claiming that the interruption of cell signals mediated by electrophiles is related to the occurrence of atherosclerosis and cancer. HNE and ONE and their derivatives are both active lipid electrophile reagents that inhibit the release of proinflammatory factors to a certain extent [144]. Nε-carboxymethyl-lysine (CML) has been reported to accumulate in large amounts in the tissues of diabetes and atherosclerosis, and glucosone aldehyde is related to its formation [145]. Benzyl isothiocyanate salt has been reported to inhibit lipid production and fatty liver formation in obese mice fed a high-fat diet [146]. It has been reported in the literature that thiazolidine derivatives have a positive effect in the treatment of LDLR(-/-) atherosclerotic mice [147]. In addition, the carboxyethylation of lysine has also changed, and some research indicates that the degree of carboxymethylation and carboxyethylation of lysine in the plasma of diabetic mice is significantly increased [148]. Changes in the expression of fucosylated oligosaccharides have been observed in pathological processes such as atherosclerosis [149]. In addition, the phosphorylation modification of tyrosine is related to the formation of esters, which may also be involved in lipid metabolism and the occurrence and progression of diseases [150].

In the differential modifications between the experimental group and the control group, some of the significantly changed modifications also changed in the comparison within the experimental group. In addition, the Delta_H(2)C(2) modification at the N-terminus of the amino acid also refers to acetaldehyde +26. In addition, acetaldehyde stimulates the growth of vascular smooth muscle cells in a notch-dependent manner, promoting the occurrence of atherosclerosis [151]. Advanced protein glycosylation is an important mechanism for the development of advanced complications of diabetes, including atherosclerosis. Hydroimidazolone-1 derived from methylglyoxal is the most abundant advanced glycosylation end-product in human plasma [152]. In addition, the guanidine modification of lysine may also be related to atherosclerosis [153].

Although not verified in the restricted search, an increasing number of studies have shown that short-chain fatty acids and their homologous acylation are involved in cardiovascular disease, and the proportions of 2-hydroxyisobutyrylation, malonylation and crotonylation in the experimental group were significantly increased [154]. Nε-carboxymethyl-lysine (CML) has been reported to accumulate in large amounts in the tissues in diabetes and atherosclerosis, and its induced PI3K/Akt signal inhibition promotes foam cell apoptosis and the progression of atherosclerosis [155]. In addition, glucosone is closely related to its formation, the proportion of which also increased significantly in the experimental group. Oxidation of tyrosine produces dihydroxyphenylalanine (DOPA), and the protein binding DOPA in tissues is elevated in many age-related pathological diseases, such as atherosclerosis and cataract formation [156].

As mentioned above, the comparison within the experimental group avoids the influence of genes, diet and other factors on the urinary proteome, but it may be affected by the growth and development of the organisms themselves. Comparison between the experimental group and control group avoids the influence of development but cannot avoid factors such as diet. The identification results of chemical modifications of urine proteins showed that regardless of whether comparison within the experimental group was adopted, the modification status changed greatly and was closely related to lipids and cardiovascular diseases. In comparison, differences between the experimental group and control group may be more obvious.

## 4. Conclusions

This study explored changes in urinary proteomics of high-fat-diet-fed *ApoE*^−/−^ mice. The results of comparison within the experimental group showed that even after only one week of a high-fat diet, while the urinary proteome of the control group had not significantly changed, the urinary proteome of the experimental group had changed significantly, and most of the enriched biological pathways were related to lipid metabolism and glycometabolism, indicating that the urinary proteome has the potential for early and sensitive monitoring of biological changes. Most of the proteins and their family members that change continually in disease progression have been reported to be related to cardiovascular diseases and/or can be used as biomarkers. The results of the comparison between the experimental group and the control group show that the biological processes enriched by differential proteins at different time points correspond to the occurrence and development of atherosclerosis, indicating that the urinary proteome has the potential to be used to monitor the disease process. The differential modification types in the comparison within the experimental group and the comparison between the experimental and control groups have also been reported to be related to lipids and cardiovascular diseases and can be used as a reference for identifying new biomarkers.

## Figures and Tables

**Figure 1 biomolecules-12-01569-f001:**
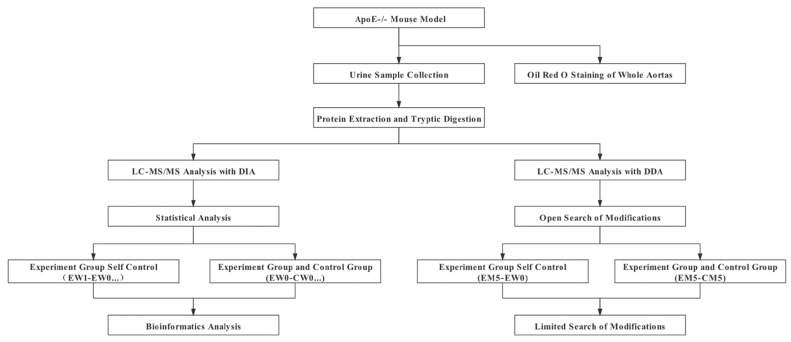
Experimental flow graph.

**Figure 2 biomolecules-12-01569-f002:**
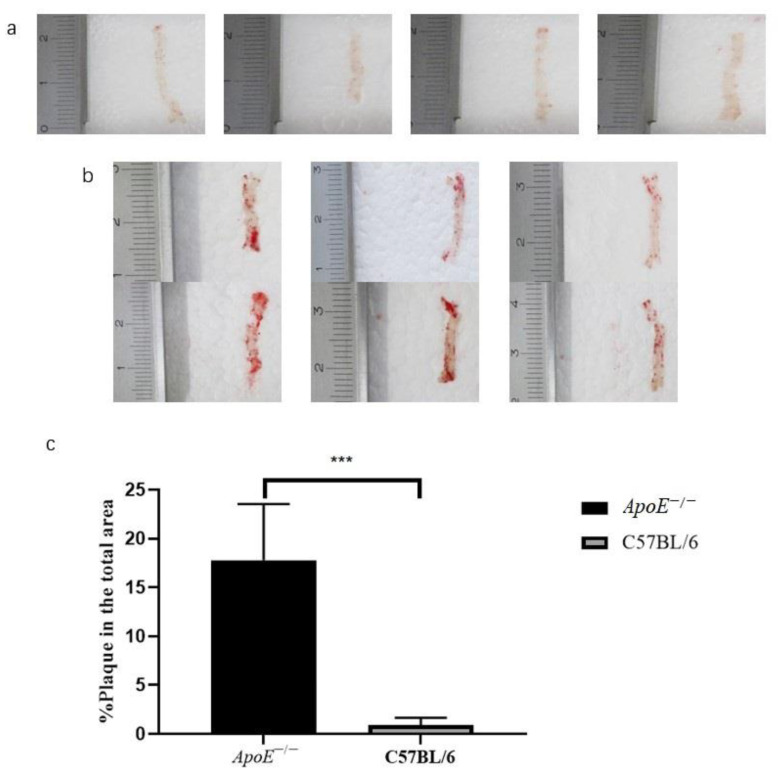
Results and quantitative analysis of oil red O staining of the whole aorta in the (**a**) control group and (**b**) experimental group, and (**c**) comparison of the staining area ratio, ***, *p* < 0.001.

**Figure 3 biomolecules-12-01569-f003:**
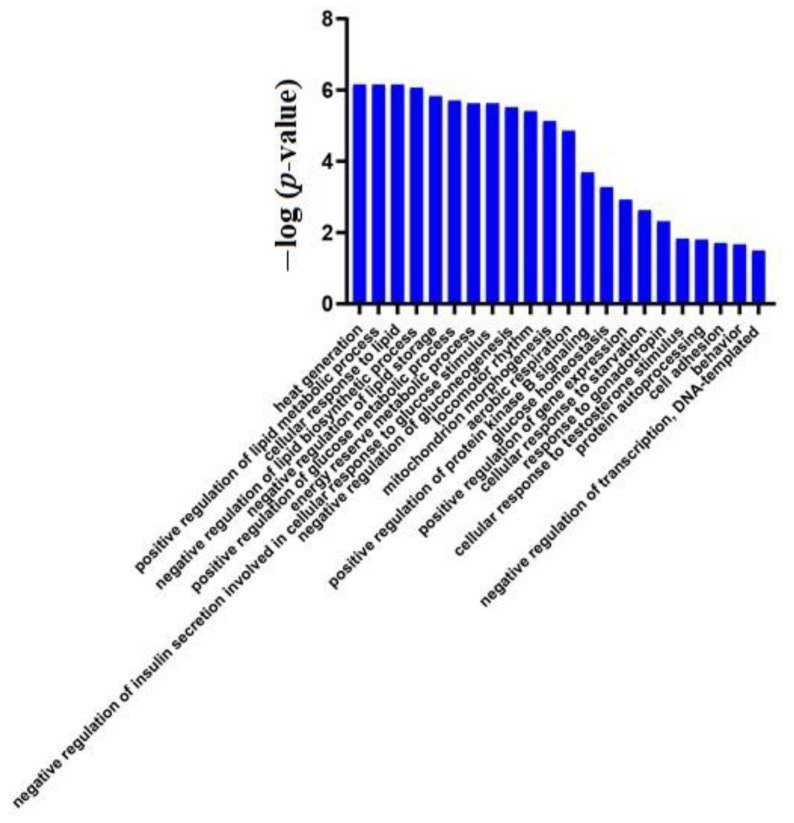
Biological processes enriched in differential proteins between week 1 and week 0 samples of the experimental group (*p <* 0.05).

**Figure 4 biomolecules-12-01569-f004:**
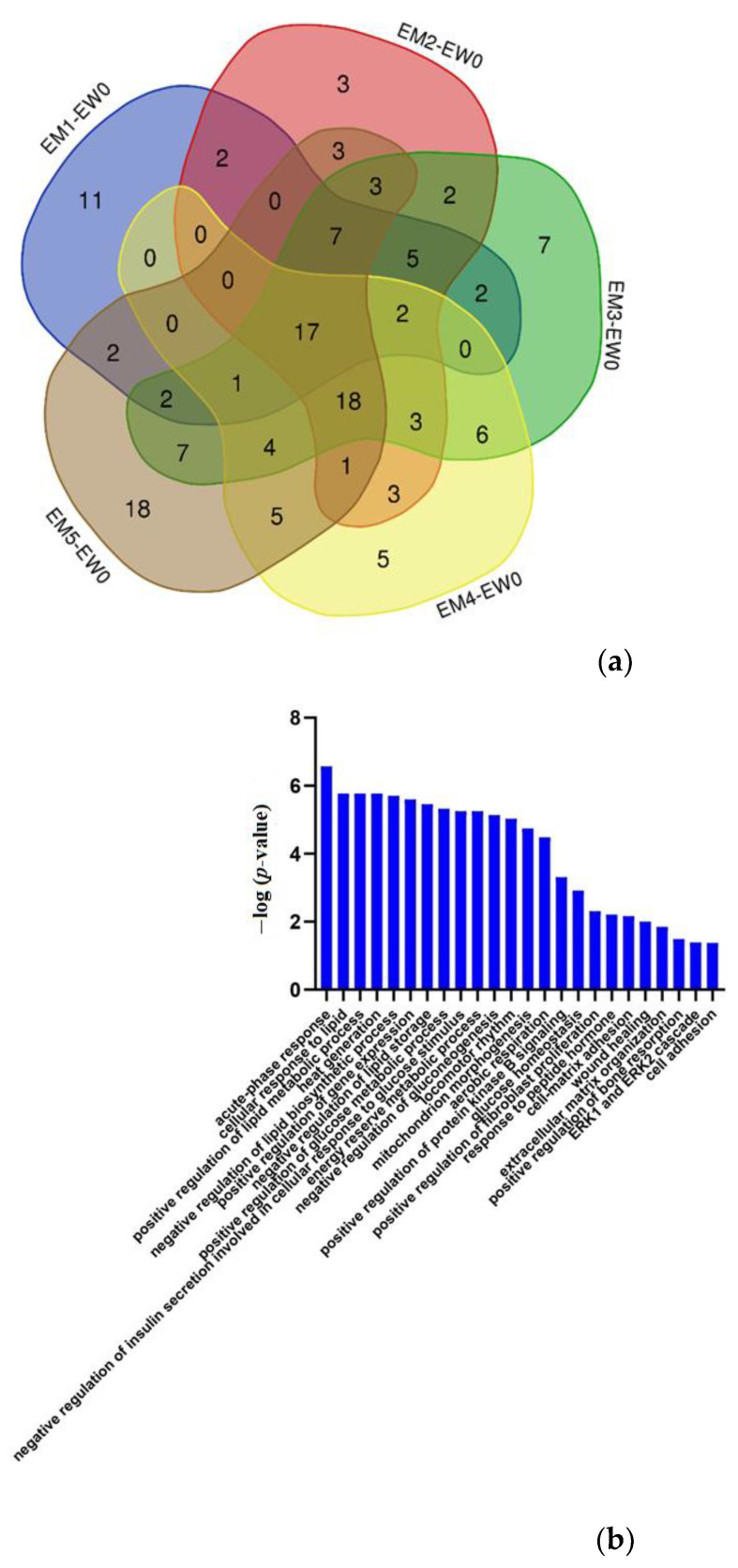
Differential proteins in the whole process. (**a**) Venn diagram of differential proteins among the other time points (M1/M2/M3/M4/M5) and week 0 samples in the experimental group. (**b**) Biological processes enriched by continuously changing proteins in the comparison within the experimental group (*p* < 0.05).

**Figure 5 biomolecules-12-01569-f005:**
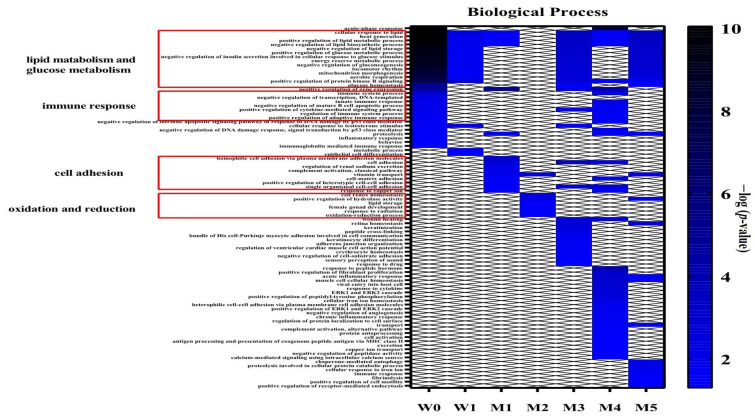
Functional annotation of differential proteins at different time points between the experimental and control groups (p < 0.05). When the experimental group is compared to the control group, there is a large difference in W0, demonstrating that the urinary proteome reflects even slight difference between the groups. In the subsequent control results at each time point, the degree of overlap in the differential proteins is small, but they are mostly related to lipids and cardiovascular diseases. The enriched biological processes also correspond to the progression of atherosclerosis, indicating that the urinary proteome is useful to monitor the disease process. However, as mentioned before, this type of comparison does not take the influence of diet and other factors into account.3.3. Chemical Modifications of Proteins.

**Figure 6 biomolecules-12-01569-f006:**
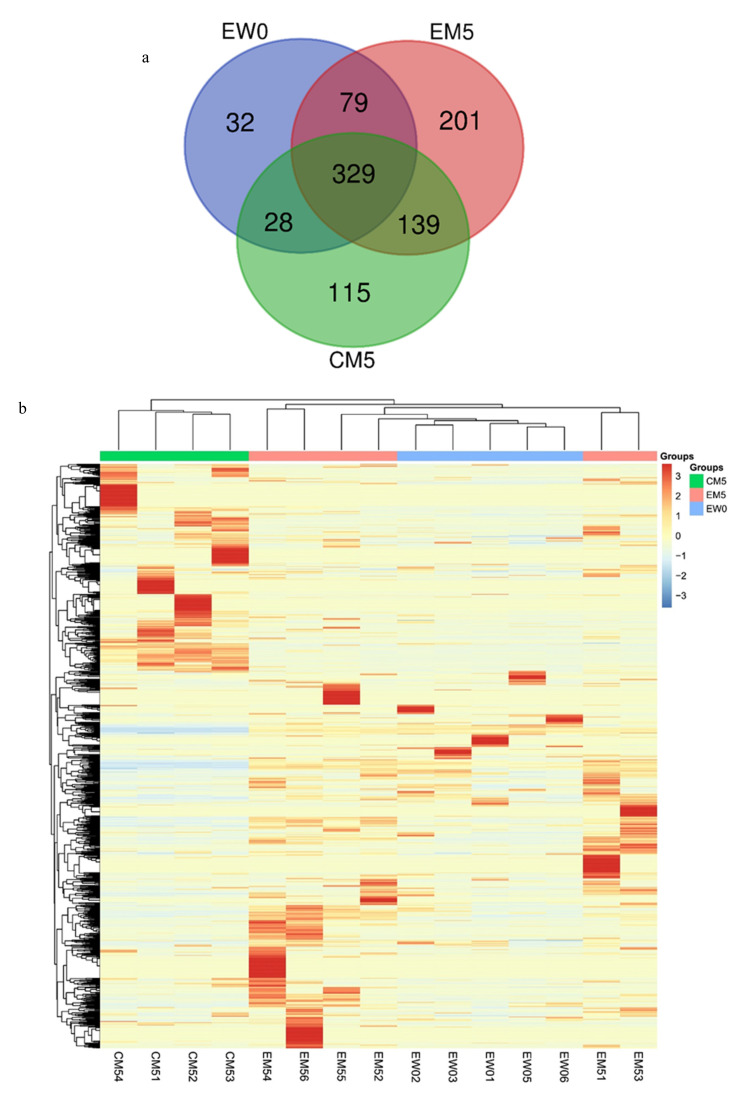
Chemical modifications among the three groups. (**a**) Venn diagram of modification types among the three groups. (**b**) Unsupervised clustering of all the modification types in the three groups. (**c**) An exemplary spectrum of modifications.

## Data Availability

The mass spectrometry proteomics data have been deposited to the ProteomeXchange Consortium (http://proteomecentral.proteomexchange.org (accessed on 10 August 2022)) via the iProX partner repository with the dataset identifier PXD027610.

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
