# Peer review of "Changes to Urinary Proteome in High-Fat-Diet ApoE−/− Mice"

_biomolecules, 2022, doi:10.3390/biom12111569_

Round 1
Reviewer 1 Report
My review of the paper was hampered by not being able to view the data deposited in iPRoX despite having the account details from biomolecules and also not having access to the supplemental data that is referred to in the text.
While the methods used are technically sound, the experimental design and data analysis could be approved upon.
The decision to compare wild-type mice on a normal diet with knock-out mice on a high-fat diet means it is not possible to infer what changes are due to the diet and what is due to the knock-out. It would be interesting to look at fewer time points but include a knock-out chow diet set which would presumably have an intermediate plaque load and this could have aided the evaluation of their putative biomarkers.
When presenting the results there is a heavy emphasis on a pair-wise comparison of select timepoints and groups. It is often unclear what statistical thresholds have been applied to the analysis and there appear to be few biomarkers that correlate with the presumable increase in plagues over the duration of the experiment. I would have liked to have seen (probably in the supplementary data) a PCA plot and heat map of the proteomics data for the experiment as a whole. For the pairwise comparisons, volcano plots are very useful. Given the multiple time points, some temporal plots of protein abundance cluster by clade showing similar profiles would have been insightful.
I think with a more thorough analysis of the data will enable them to focus on the most interesting potential biomarkers, which in turn would lead to a more focused discussion in the paper.
For the modification data, including annotated spectra in the supplemental data of all the modifications that make it through the analysis would be of help to the reader.
Author Response
Dear professor, thank you for your nice comments.
For the iPRoX item, according to the instructions of the website, since the article has not been accepted, the dataset is not yet publicly released. So we share a linkage for you to make it accessible. The URL is https://www.iprox.cn/page/SSV024.html;url=1664608671157UxDb and the password is AvWJ with a period of validity of 360 days. If you have any further questions, please feel free to contact us.
And the supplemental data has also been reloaded.
For the experimental design and data analysis items, we have the following response:
- The decision to compare wild-type mice on a normal diet with knock-out mice on a high-fat diet means it is not possible to infer what changes are due to the diet and what is due to the knock-out. It would be interesting to look at fewer time points but include a knock-out chow diet set which would presumably have an intermediate plaque load and this could have aided the evaluation of their putative biomarkers.
We agree with you it is important to see what changes are due to the diet and what is due to the knock-out. In fact, we took both factors into account in the experimental design. For example, we believe that in the experimental group, from week 0 to week 1, the influence of other factors such as age and plaque is rather minimal, thus when we compare the urine samples of EW0 (week 0 of experimental group) and EW1, changes in the urinary proteome are mostly due to the high-fat-diet and they are related to lipid metabolism and glucose metabolism (see Section 3.2.1(1)). It is similar to the comparison between EW0 and CW0 samples. Before the feed of different diets, knock-out is the only variable, and the urinary proteome can tell us there are already significant differences in the biological processes related to lipid and glycometabolism (see Section 3.2.2(1)). It would be better if we can establish a lot of groups under different conditions to explore the specific effect of each factor, however, for the purpose of early diagnosis and biomarker detection, given that atherosclerosis is a mixture of many factors (Libby P, Buring JE, Badimon L, Hansson GK, Deanfield J, Bittencourt MS, TokgözoÄŸlu L, Lewis EF. Atherosclerosis. Nat Rev Dis Primers. 2019 Aug 16;5(1):56. doi: 10.1038/s41572-019-0106-z. PMID: 31420554.), our study can only provide clues in the search for biomarkers at current stage.
- When presenting the results there is a heavy emphasis on a pair-wise comparison of select timepoints and groups. It is often unclear what statistical thresholds have been applied to the analysis and there appear to be few biomarkers that correlate with the presumable increase in plagues over the duration of the experiment. I would have liked to have seen (probably in the supplementary data) a PCA plot and heat map of the proteomics data for the experiment as a whole. For the pairwise comparisons, volcano plots are very useful. Given the multiple time points, some temporal plots of protein abundance cluster by clade showing similar profiles would have been insightful.
I think with a more thorough analysis of the data will enable them to focus on the most interesting potential biomarkers, which in turn would lead to a more focused discussion in the paper.
When it comes to the pair-wise comparison, since urine is so sensitive (Wei J, Gao Y. Early disease biomarkers can be found using animal models urine proteomics. Expert Rev Proteomics. 2021 May;18(5):363-378. doi: 10.1080/14789450.2021.1937133. Epub 2021 Jun 7. PMID: 34058951.), it is essential to minimize the influence of other factors, otherwise, the influence of experimental factors may be masked. For instance, in the heat map added to the supplemental material, we can find that it is hard to discriminate samples of different time points or groups as a whole, which indicates that there are great differences among individuals. As a result, we set up a comparison within the experimental group to avoid the influence of genetic and dietary differences as well as a comparison between the experimental group and control group to offset the influence of biological growth and development. Besides, when we try to validate these proteins in clinical research, pair-wise comparison is even better than group comparison in personalized medicine in the future Volcano plots have been added to show the changes in pair-wise comparison more specifically as well as statistical thresholds. However, we hold the view that it is not necessary to seek biomarkers over the duration of the experiment. There are several stages in the development of atherosclerosis (Libby P, Buring JE, Badimon L, Hansson GK, Deanfield J, Bittencourt MS, TokgözoÄŸlu L, Lewis EF. Atherosclerosis. Nat Rev Dis Primers. 2019 Aug 16;5(1):56. doi: 10.1038/s41572-019-0106-z. PMID: 31420554.), as we can see from Figure 5, different biological processes were enriched by GO analysis at different time points. It is unlikely for the same proteins to be able to reflect the whole disease progression.
- For the modification data, including annotated spectra in the supplemental data of all the modifications that make it through the analysis would be of help to the reader.
For the modification data, an exemplary spectrum was added to Figure 6. PFind Studio software (version 3.1.6, Institute of Computing Technology, Chinese Academy of Sciences) was used and modifications were identified and calculated.
We sincerely hope this reply is helpful in solving your problems, thanks a lot for your review.
Reviewer 2 Report
Dear authors,
After analysis of your manuscript I consider that a few issues should be addressed:
1. Abstract: I do not understand the meaning of the sentence “Urine ... is not regulated by homeostasis mechanisms...”. Please rewrite.
I’m not sure I understood what you mean by the expressions “Internal control” and “Intergroup control”. I don't know if these are the most appropriate ones.
2. Introduction: The novelty of the study should be referred.
References are missing in some parts, such as at the end of the sentences: “The impact of atherosclerosis is more significant during the late stages and induces a series of fatal consequences, such as myocardial infarction and stroke.”, and “In addition, urine is more accessible and non-invasive to obtain.”
3. Results: In tables 1 and 2 what do the references mean?
Also, at the end of table 2 (page 17) is there information missing regarding Uniprot, Fold Change, and References?
4. Discussion: In the Discussion section, also some references are missing. Please revise the entire manuscript and add the missing references.
5. What about the validation of some of the proteins identified using, for instance, slot or western blotting?
Author Response
Dear professor, thank you for your nice comments.
We apologize for your confusion.
- Abstract:
The abstract has been amended to “Urine bears no need nor mechanism to be stable, so it accumulates many small changes, is therefore a good source of biomarkers in the early stage of disease”, what we really mean is any change that is introduced into the blood either internally or externally tends to be cleared by the liver, kidney and/or other organs via a variety of mechanisms in order to maintain the homeostasis of the blood. In contrast, urine is the place that most of the wastes in the blood are dumped into, and thus tolerates changes to a much higher degree.
The “comparison within the experimental group” and “comparison between the experimental group and control group” are used to replace “internal control” and “intergroup control”. The comparison within the experimental group avoids the influence of genetic and dietary differences on the experimental results to the greatest extent, and the comparison between the experimental group and the control group excludes the influence of biological growth and development.
- Introduction:
The novelty of this study is emphasized in the amended introduction, which is “Although the early diagnosis of atherosclerosis is important and a large number of biomarkers have been identified, few works are involved in the urinary biomarkers. Besides, urine is an ideal source of early biomarkers and has the potential to reflect the changes in the early stages of atherosclerosis. Therefore, there is an urgent need to explore the changes of the urinary proteome in high-fat diet ApoE-/- mice.”
The missing references were also added.
- Results:
The references in Tables 1 and 2 show that the protein has been reported to be related to cardiovascular diseases or lipid metabolism, indicating that the results of our study are strongly supported by others and urine has the potential to provide clues in the search for biomarkers.
The missing information at the end of table 2 is due to the format adaption. Now the table was readjusted and we wish this question can be solved.
- Discussion:
The entire manuscript was revised and the missing references were added.
For the technique “liquid chromatography coupled with tandem mass spectrometry” used in the study, proteins were quantitated by summing all fragments, whereas other antibody methods such as the western blot were by epitope recognition focusing on the protein bands with a molecular weight consistent with the candidate theoretical whole protein. The principles of the two experimental methods are completely different. The quality of mass spectrometry data is vastly superior for several reasons (Aebersold R, Burlingame AL, Bradshaw RA. Western blots versus selected reaction monitoring assays: time to turn the tables? Mol Cell Proteomics. 2013 Sep;12(9):2381-2. doi: 10.1074/mcp.E113.031658. Epub 2013 Jun 10. PMID: 23756428; PMCID: PMC3769317.). (i) Quality of the assay: Quantification by Western blotting is based on a single reagent (antibody) that may be poorly characterized. Not uncommonly, neither its affinity for the antigen nor the epitope is known or disclosed. Further, frequently no bona fide reference sample is available to test the performance of the assay in the context of a particular experiment. (ii) Quality of the results: Protein quantification via Western blotting depends on a single signal: the intensity of a band on the blot. This signal can be specific (i.e. represent the targeted protein) or unspecific. (iii) Performance characteristics: Each method is characterized by a number of performance characteristics such as limit of detection, linear dynamic range, ability to multiplex, and reproducibility. For most of these characteristics, mass spectrometry-based methods now outperform.
As a result, it is not a common practice to validate the mass spectrometry results in the antibody methods in recent years. And for the high-throughput mass spectrometry technique, more disease clues can be provided.
We sincerely hope this reply is helpful in solving your problems, thanks a lot for your review.
Round 2
Reviewer 2 Report
The authors have well addressed all of my concerns.